# Combating Noisy Labels via Dynamic Connection Masking

## Abstract

Noisy labels are inevitable in real-world scenarios. Due to the strong capacity of deep neural networks to memorize corrupted labels, these noisy labels can cause significant performance degradation. Existing research on mitigating the negative effects of noisy labels has mainly focused on robust loss functions and sample selection, with comparatively limited exploration of regularization in model architecture. In this paper, we propose a **Dynamic Connection Masking (DCM)** mechanism for the widely-used Fully Connected Layer (FC) to enhance the robustness of classifiers against noisy labels. The mechanism can adaptively mask less important edges during training by evaluating their information-carrying capacity. Through this selective masking process of preserving only a few critical edges for information propagation, our DCM effectively reduces the gradient error caused by noisy labels. It can be seamlessly integrated into various noise-robust training methods to build more robust deep networks, including robust loss functions and sample selection strategies. Additionally, we validate the applicability of our DCM by extending it to the newly-emerged Kolmogorov-Arnold Network (KAN) architecture. The experimental results reveal that the KAN exhibits superior noise robustness over FC-based classifiers in real-world noisy scenarios. Extensive experiments on both synthetic and real-world benchmarks demonstrate that our method consistently outperforms state-of-the-art (SOTA) approaches. Code is available at https://anonymous.4open.science/r/DCM-0C0A.

## 1 Introduction

Deep neural networks (DNNs) have achieved remarkable performance in various supervised classification (Rawat & Wang, 2017; Abdou, 2022; Evans et al., 2022; Guo et al., 2023). The success largely depends on large-scale, accurately labeled data. However, acquiring high-quality labeled data remains prohibitively expensive in practice, inevitably introducing noisy labels into training datasets. Extensive studies have shown that training with these corrupted labels can cause significant performance degradation, as DNNs are prone to overfitting on corrupted labels (Zhang et al., 2021; Johnson & Khoshgoftaar, 2022; Qian et al., 2023). Consequently, robust learning with noisy labels has become a critical research focus in deep learning.

Existing noise-robust training methods primarily focus on robust loss functions and sample selection strategies (Ghosh et al., 2017; Song et al., 2019a; Sun et al., 2020; Gao et al., 2021; Liu et al., 2024a). The former achieves risk minimization by optimizing the loss function, which particularly requires multiple parameters to balance between noise tolerance for mislabeled samples and sufficient learning for clean samples (Wang et al., 2021; Chen et al., 2025). The latter seeks to identify true labeled examples for training, which relies on various heuristic criteria (*e.g.*, small loss (Jiang et al., 2018; Shen & Sanghavi, 2019), predicted probability (Yi & Wu, 2019; Sheng et al., 2024)). Additionally, some popular regularization techniques can also mitigate overfitting to noisy data, such as Dropout (Srivastava et al., 2014) and DropConnect (Wan et al., 2013). By randomly discarding neurons or connections of the Fully Connected Layer (FC), they implicitly average over an ensemble of subnetworks and reduce overfitting. Nonetheless, their inherent randomness means that they are not specifically tailored for suppressing noisy information, making it difficult to balance the propagation of noisy and clean signals. Motivated by this limitation, as illustrated in Figure 1, we aim

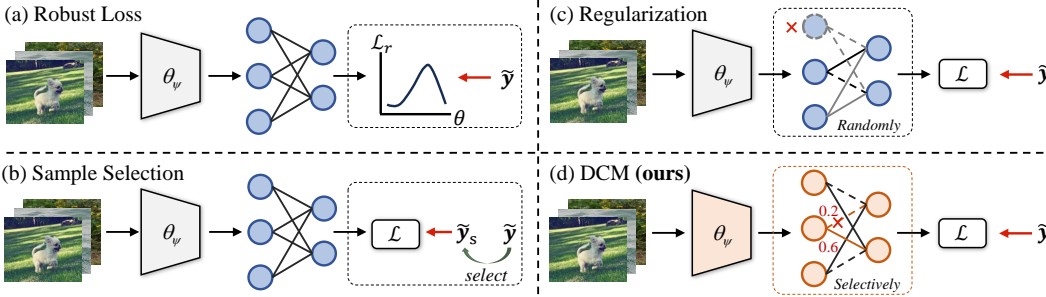

Figure 1: Comparison of various methods for learning with noisy labels. Robust loss functions achieve noise-tolerant loss for optimization. Sample selection strategies aim to identify clean data $\tilde{y}_s$ from noisy samples $\tilde{y}$. Popular regularization methods, such as Dropout or DropConnect, randomly remove neurons or connections to mitigate overfitting. Our DCM selectively adjusts the classifier connections, allowing only important pathways for gradient backpropagation.

to combat noisy labels through simple architectural regularization, which effectively mitigates the propagation of noisy gradient without degrading the clean information.

To this end, we propose a novel **Dynamic Connection Masking** (DCM) mechanism for the widely-used FC to enhance the robustness of the classifier against noisy labels. Intuitively, the negative impacts of noisy labels arise from gradient backpropagation during training. Reducing these noise-contaminated gradients would straightforwardly mitigate the adverse effects. Therefore, our DCM dynamically masks less important edges by evaluating their information-carrying capacity. If an edge carries less information, it would contribute less to learning but have the risk of backpropagating noisy gradients. Consequently, temporarily discarding them in each training step can suppress gradient errors without damaging the information propagation in the network. By operating intrinsically within the network architecture, our DCM is orthogonal to existing methods that act externally, such as robust loss functions (loss-level) and sample selection strategies (data-level). Consequently, it can be seamlessly integrated with these methods as a plug-and-play module to achieve further performance enhancement. Additionally, to further validate the effectiveness of our DCM, we apply it to the newly proposed Kolmogorov–Arnold Network (KAN) (Liu et al., 2024b) architecture. Interestingly, we find that KAN exhibits superior noise robustness on real-world datasets compared with FC-based classifiers. The main contributions of this paper are summarized as:

- We propose a novel dynamic connection masking mechanism for both widely-used FC-based and newly-emerged KAN-based classifiers for learning with noisy labels. Through adaptive edge masking during training, the approach effectively reduces the gradient error caused by noisy labels while simultaneously maintaining its capacity to fit clean data.

- We integrate our approach into existing noise-robust training methods, including robust loss functions and sample selection strategies. Evaluations on both synthetic and real-world datasets demonstrate the superiority of our approach, achieving SOTA performance.

- To the best of our knowledge, this is also the first work to extend the applicability of KAN to learning from noisy labels in classification tasks. Experimental results demonstrate that KAN exhibits enhanced robustness to label noise over FC in real-world scenarios.

## 2 RELATED WORK

**Robust Loss Function.** Robust loss design has been extensively studied (Qin et al., 2019; Feng et al., 2021; Sztukiewicz et al., 2024; Wilton & Ye, 2024). Theoretical studies have shown that certain losses like Mean Absolute Error (MAE) possess inherent noise robustness (Ghosh et al., 2017). However, empirical results indicate that MAE converges slowly (Zhang & Sabuncu, 2018). Beyond this observation, the generalized cross-entropy (GCE) loss (Zhang & Sabuncu, 2018) combines MAE's robustness with CCE's efficiency via Box-Cox transformation, allowing fast training and noise tolerance. Further studies include Active Passive Loss (APL) (Ma et al., 2020), which normalizes arbitrary losses into robust forms via active-passive combining. Furthermore, Sparse Regu-

larization (SR) (Zhou et al., 2021b) imposes the $\ell_p$-norm constraint into the loss function for robust training. Recently, the Active Negative Loss (ANL) (Ye et al., 2024) enhances APL by incorporating Normalized Negative Loss Functions, proposing a novel framework for improved performance.

**Sample Selection.** Unlike loss optimization, sample selection strategies aim to identify correctly labeled examples from noisy data through multi-network or multi-round learning (Yu et al., 2019; Shen & Sanghavi, 2019; Patel & Sastry, 2023). For instance, Co-teaching (Han et al., 2018) employs two parallel networks that cross-update using small-loss samples selected from each other, thereby reducing error accumulation. Jo-SRC (Yao et al., 2021) employs Jensen-Shannon divergence to assess prediction consistency across augmentations. DISC (Li et al., 2023) dynamically adjusts instance-specific thresholds based on its memorization momentum across training epochs, enabling adaptive noise correction. Recently, SED (Sheng et al., 2024) introduces class-balanced selection via adaptive probability thresholds, improving robustness under class-imbalanced noise.

**Regularization.** Regularization techniques enhance model generalization by imposing constraints on the model. Widely adopted methods include Dropout (Srivastava et al., 2014) and DropConnect (Wan et al., 2013), which randomly disable hidden units or mask individual connections via Bernoulli sampling. However, these popular regularization methods are not tailored for noisy label scenarios and often exhibit suboptimal performance under high noise levels (Song et al., 2022). Consequently, advanced regularization methods have been proposed, such as early stopping (Rolnick et al., 2017; Song et al., 2019b; Li et al., 2020). Among these, CDR (Xia et al., 2020) identifies critical parameters via gradient-weight products and penalizes only the noncritical ones to suppress their influence. However, gradient-based paramter screening in CDR incurs additional computational overhead. Therefore, we aim to implement simple and efficient parameter selection to enable stable and robust learning during training.

**Kolmogorov-Arnold Networks.** Inspired by the Kolmogorov-Arnold representation theorem, KAN (Liu et al., 2024b) serves as a promising alternative to the traditional Multi-Layer Perceptron Network (MLP). Unlike MLP with fixed activation functions at nodes, KAN utilizes learnable activation functions on edges. Specifically, each weight parameter is modeled as a univariate function, typically parameterized by spline functions. This architecture enhances model flexibility to better adapt to diverse data patterns (Somvanshi et al., 2024; Mohan et al., 2024). While KAN has demonstrated effectiveness in various machine learning tasks (Cheon, 2024; Vaca-Rubio et al., 2024; Ji et al., 2024), its robustness to noisy labels remains underexplored.

## 3 METHOD

### 3.1 PRELIMINARIES

Consider a single-label classification problem with a total number of $C$ classes. In an ideal scenario, let $D = \{(x_i, y_i)\}_{i=1}^N$ denote a clean training set, where $x_i$ represents the $i$-th training image, and $y_i = \{0, 1\}^C$ indicates its one-hot encoded true label. However, acquiring a perfectly clean dataset with accurate labels $y_i$ is often impractical. Instead, we typically have access to a noisy dataset $D_\eta = \{(x_i, \tilde{y}_i)\}_{i=1}^N$, where $\tilde{y}_i$ represents the observed label that may differ from the true label.

A general classification model $f$ consists of two components, which can be expressed as $f = g \circ \psi$, where a visual backbone $\psi$ extracts feature maps for the input image $x_i$, and a classifier $g$ projects the input feature space to a probability distribution over the label space. The training objective is to encourage that the global minimizer $f^*$ obtained in the presence of label noise also serves as the global minimizer under clean label supervision (Zhang & Sabuncu, 2018).

### 3.2 DYNAMIC CONNECTION MASKING

As illustrated in Figure 2, our approach encompasses two key processes: (i) edge importance scoring and (ii) edge masking. Specifically, we first compute the importance score for each connection, quantifying its ability to transmit information. Then, we dynamically mask edges with lower importance scores during training. Our approach enables the network to automatically adjust its connectivity pattern, maintaining only the most informative pathways while suppressing potentially misleading signals from noisy labels.

Figure 2: Overview of our dynamic connection masking mechanism. (i) We first compute the edge activation value $\boldsymbol{A} \in \mathbb{R}^{B \times C \times d}$ via multiplication between the input feature $v_{ik}$ and its corresponding edge weight $w_{jk}$, where $B$, $C$ and $d$ denote batch size, total class number, and the dimension of the input feature. Then, the edge importance score $\boldsymbol{S}$ is obtained by measuring the standard deviation of $\boldsymbol{A}$ along the batch dimension (Eq. 2). (ii) We adaptively mask edges with lower importance scores during training, dynamically adjusting the masking of connections at each timestep interval $t$.

### 3.2.1 EDGE IMPORTANCE SCORING

Intuitively, the importance of an edge corresponds to its ability to convey information. Specifically, edges transmitting more information inherently possess greater significance. To quantify each edge's information-carrying capability, we adopt the standard deviation to measure the dynamic activation variability of each edge during forward propagation across different samples. A larger variance of an edge indicates that it carries more discriminative information and thus exhibits more importance.

Given the input features $\boldsymbol{v} \in \mathbb{R}^{B \times d}$ extracted by the visual backbone $\psi$, and the learnable weight matrix $\boldsymbol{W} \in \mathbb{R}^{C \times d}$ in a single-layer classifier, the activation value of each edge $\boldsymbol{A} \in \mathbb{R}^{B \times C \times d}$ is obtained by multiplying the input feature with the corresponding edge weight:

$$a_{ijk} = v_{ik} \times w_{jk}, \tag{1}$$

where $B$ is the batch size, $i \in \{1, ..., B\}$, $j \in \{1, ..., C\}$, and $k \in \{1, ..., d\}$ represent the index of the sample, output node, and input node, respectively. Subsequently, the edge importance score $\boldsymbol{S} \in \mathbb{R}^{C \times d}$ is defined as the standard deviation of the edge activation value $\boldsymbol{A}$ across samples:

$$s_{jk} = \sqrt{\frac{1}{B} \sum_{i=1}^{B} (a_{ijk} - \mu_{jk})^2}, \quad \mu_{jk} = \frac{1}{B} \sum_{i=1}^{B} a_{ijk}, \tag{2}$$

where $\mu_{jk}$ denotes the mean activation value of the edge between the output node $j$ and input node $k$ across all samples. Our edge scoring mechanism evaluates the importance of individual edges, thereby establishing an optimized basis for dynamic edge masking.

### 3.2.2 EDGE MASKING

The dynamic masking mechanism adaptively adjusts the classifier connectivity using edge importance scores. Specifically, it involves updating the mask matrix to control which edges are retained or temporarily discarded. We introduce a hyperparameter $p \in (0, 1)$ to control the masking ratio. For each input node $k$, its connections are ranked in ascending order by importance scores $s_{jk}$, and the bottom $q = \lfloor p \times C \rfloor$ of these connections are removed, where $\lfloor \cdot \rfloor$ denotes the floor operation.

Formally, the masking edges set $U_k$ contains indices of edges to be masked for each input node $k$:

$$U_k = \left\{ j \mid j \in \underset{j}{\operatorname{argsort}} (s_{jk})_{1:q} \right\}, \tag{3}$$

where $\operatorname{argsort}(\cdot)$ returns the edge indices $j$ of $s_{jk}$ sorted in ascending order. Then, the binary mask matrix $\boldsymbol{M} \in \mathbb{R}^{d \times C}$ is defined as:

$$m_{kj} = \begin{cases} 0, & j \in U_k \\ 1, & \text{otherwise} \end{cases}, \tag{4}$$

Table 1: Comparison with SOTA robust loss function methods on CIFAR-10 and CIFAR-100 datasets under various noise rates. Results of existing methods are mainly drawn from APL (Ma et al., 2020). The results (mean±std) are reported over 3 random runs, and the top 2 best results are highlighted using boldface and underlining. The blue-highlighted regions represent the best method.

| Datasets | Methods | Sym-20% | Sym-40% | Sym-60% | Sym-80% | Asym-20% | Asym-40% |
|---|---|---|---|---|---|---|---|
| CIFAR-10 | GCE | 87.27±0.21 | 83.33±0.39 | 72.00±0.37 | 29.08±0.80 | 86.07±0.31 | 74.98±0.32 |
| | NLNL | 83.98±0.18 | 76.58±0.44 | 72.85±0.39 | 51.41±0.85 | 84.74±0.08 | 76.97±0.52 |
| | SCE | 88.05±0.26 | 82.06±0.24 | 66.08±0.25 | 30.69±0.63 | 83.92±0.07 | 78.20±0.03 |
| | SR | 87.93±0.07 | 84.86±0.18 | 78.18±0.36 | 51.13±0.51 | 87.70±0.19 | 79.29±0.20 |
| | APL | 89.22±0.27 | 86.02±0.09 | 79.78±0.50 | 52.71±1.90 | 88.56±0.17 | 79.59±0.40 |
| | ANL | 89.72±0.04 | 87.28±0.02 | 81.12±0.30 | 61.27±0.55 | 89.13±0.11 | 77.63±0.31 |
| | APL-DFC | 89.34±0.23 | 86.26±0.06 | 80.32±0.15 | 56.99±1.52 | 88.84±0.15 | 80.14±0.21 |
| | APL-DKAN | 89.60±0.24 | 86.49±0.16 | 80.25±0.21 | 54.39±0.48 | 88.71±0.53 | 80.63±0.15 |
| | **ANL-DFC** | 89.93±0.13 | **87.45±0.05** | **81.80±0.20** | 62.98±0.95 | **89.56±0.27** | 81.05±0.29 |
| | **ANL-DKAN** | **90.16±0.02** | 87.32±0.09 | 81.69±0.19 | **63.49±0.09** | 89.37±0.18 | **81.47±0.30** |
| CIFAR-100 | GCE | 65.24±0.56 | 58.94±0.50 | 45.18±0.93 | 16.18±0.46 | 59.99±0.83 | 41.49±0.79 |
| | NLNL | 46.99±0.91 | 30.29±1.64 | 16.60±0.90 | 11.01±2.48 | 50.19±0.56 | 35.10±0.20 |
| | SCE | 55.39±0.18 | 39.99±0.59 | 22.35±0.65 | 7.57±0.28 | 58.22±0.47 | 42.19±0.19 |
| | SR | 67.51±0.29 | 60.70±0.25 | 44.95±0.65 | 17.35±0.13 | 64.79±0.01 | 49.51±0.59 |
| | APL | 65.31±0.07 | 59.48±0.56 | 47.12±0.62 | 25.80±1.12 | 62.68±0.79 | 46.79±0.96 |
| | ANL | 67.09±0.32 | 61.80±0.50 | 51.52±0.53 | 28.07±0.28 | 66.27±0.19 | 45.41±0.68 |
| | APL-DFC | 65.99±0.31 | 59.79±0.26 | 47.40±0.24 | 26.40±0.43 | 63.23±0.45 | 48.11±0.36 |
| | APL-DKAN | 66.05±0.12 | 59.66±0.19 | 48.69±0.16 | 25.98±0.05 | 64.01±0.37 | 48.35±0.40 |
| | **ANL-DFC** | 67.63±0.12 | 62.54±0.39 | 52.30±0.51 | **29.43±0.75** | **66.62±0.20** | 46.72±0.29 |
| | **ANL-DKAN** | **67.89±0.23** | **63.02±0.35** | **53.02±1.13** | 28.79±0.57 | 66.38±0.19 | **49.67±0.90** |

where $m_{kj} = 1$ indicates that the given connection is retained, otherwise discarded. During training, the mask matrix $\boldsymbol{M}^{(t)}$ is dynamically updated at each timestep interval $t$. It allows the network to continuously evolve its connectivity pattern, facilitating adaptive masking of less important edges. After applying $\boldsymbol{M}^{(t)}$, the masked weight matrix is given by $\bar{\boldsymbol{W}}^{(t)} = \boldsymbol{M}^{\mathrm{T}(t)} \odot \boldsymbol{W}^{(t)}$, where $\odot$ denotes element-wise multiplication. Then, the $\bar{\boldsymbol{W}}^{(t)}$ can be used for standard training. Benefiting from this simple masking operation, our method enables seamless integration with existing methods.

## 4 EXPERIMENTS

We implement our DCM for both FC and KAN classifiers, denoted as **DFC** and **DKAN**, respectively.

### 4.1 EXPERIMENT SETUP

**Synthetically Corrupted Datasets.** CIFAR-10 and CIFAR-100 contain $50,000$ training images and $10,000$ test images. The open-set dataset CIFAR80-NO is derived from CIFAR-100 (Krizhevsky et al., 2009), with the last 20 categories treated as out-of-distribution samples. The corrupted datasets are generated with both symmetric and asymmetric noise with noise rate $\eta \in (0, 1)$.

**Real-World Datasets.** The WebVision-Mini comprises the first 50 classes from WebVision1.0 (Li et al., 2017) for training while using the validation set as the test set. Clothing1M (Xiao et al., 2015) is a large-scale, real-world noisy dataset across 14 categories of online-crawled clothing images, with 1 million training images and $10,000$ test images.

**Compared Methods.** We evaluate our DCM by integrating it into two noise-robust training approaches: robust loss functions (APL (Ma et al., 2020), and ANL (Ye et al., 2024)), sample selection strategies (DISC (Li et al., 2023) and SED (Sheng et al., 2024)), and a hybrid method (SURE (Li et al., 2024)). Furthermore, we compare our method with several regularization methods, including Dropout (Srivastava et al., 2014), DropConnect (Wan et al., 2013) and CDR (Xia et al., 2020).

**Implementation Details.** When combining with robust loss functions, following (Ma et al., 2020; Zhou et al., 2021a), we use an 8-layer CNN for CIFAR-10 and ResNet-34 for CIFAR-100. For

Table 2: Comparison with SOTA sample selection strategies on CIFAR100 and CIFAR80N-O datasets under various noise rates. Results of existing methods are mainly drawn from SED (Sheng et al., 2024). The average test accuracy (%) is reported over the last 10 epochs, and the top 2 best results are highlighted using boldface and underlining, respectively.

| Methods | Publication | CIFAR100 | | | CIFAR80N-O | | |
|---|---|---|---|---|---|---|---|
| | | Sym-20% | Sym-80% | Asym-40% | Sym-20% | Sym-80% | Asym-40% |
| Co-teaching | NeurIPS 2018 | 43.73 | 15.15 | 28.35 | 60.38 | 16.59 | 42.42 |
| Co-teaching+ | ICML 2019 | 49.27 | 13.44 | 33.62 | 53.97 | 12.29 | 43.01 |
| JoCoR | CVPR 2020 | 53.01 | 15.49 | 32.70 | 59.99 | 12.85 | 39.37 |
| Jo-SRC | CVPR 2021 | 58.15 | 23.80 | 38.52 | 65.83 | 29.76 | 53.03 |
| Co-LDL | TMM 2022 | 59.73 | 25.12 | 52.28 | 58.81 | 24.22 | 50.69 |
| UNICON | CVPR 2022 | 55.10 | 31.49 | 49.90 | 54.50 | 36.75 | 51.50 |
| SPRL | PR 2023 | 57.04 | 28.61 | 49.38 | 47.90 | 22.25 | 40.86 |
| DISC | CVPR 2023 | 60.28 | 33.90 | 50.56 | 50.33 | 38.23 | 47.63 |
| SED | ECCV 2024 | 66.50 | 38.15 | 58.29 | 69.10 | 42.57 | 60.87 |
| DISC-DFC | | 64.18 | 35.81 | 56.25 | 60.65 | 39.79 | 51.58 |
| DISC-DKAN | Ours | 66.12 | 38.02 | 56.66 | 61.06 | 41.28 | 54.01 |
| **SED-DFC** | | 66.83 | 39.18 | **59.39** | **69.37** | **44.97** | 61.70 |
| **SED-DKAN** | | **67.16** | **39.49** | 58.75 | 69.22 | 43.08 | **62.29** |

sample selection strategies, following (Yao et al., 2021; Sheng et al., 2024), we adopt a 7-layer CNN for CIFAR-100 and CIFAR80-NO, InceptionResNetV2 (Szegedy et al., 2017) for WebVision-Mini, and ResNet-50 (He et al., 2016) for Clothing1M. When compared with regularization methods, we adhered to the optimal parameter setting from their original papers. Specifically, for both Dropout and DropConnect, we employ a random dropping rate of $0.5$. To strike a balance between noise robustness and effective learning, we select $p = 0.6$ for our DCM across all datasets and noise conditions. A detailed parameter analysis supporting this choice is provided in the Supplementary Material. All the experiments are implemented on one NVIDIA RTX-3090 GPU. More training details are also given in the Supplementary Material.

## 4.2 EVALUATION ON SYNTHETIC DATASETS

**Integration with Robust Loss Function Methods.** We integrate DCM with SOTA loss functions (APL and ANL). Experimental results on CIFAR-10 and CIFAR-100 under both symmetric and asymmetric noise are presented in Table 1. As can be observed, our method consistently achieves significant improvements, particularly as noise levels increase. For instance, under 40% asymmetric noise on CIFAR-10, our ANL-DFC and ANL-DKAN outperform the SOTA method (77.63% of ANL) by **3.42%** and **3.84%**, respectively. Overall, our DCM effectively enhances noise-robustness across different noise types and rates when integrating with existing robust loss functions.

**Integration with Sample Selection Strategies.** We integrate DCM with SOTA sample selection strategies (DISC and SED) and evaluate on both closed-set and open-set benchmarks. As demonstrated in Table 2, our methods exhibit superior robustness compared to their baseline counterparts. For example, our DISC-DKAN achieves 61.06% on CIFAR80N-O with 20% symmetric noise, surpassing DISC by **10.73%**. Notably, our SED-DFC and SED-DKAN achieve the top-2 performance rankings, establishing new SOTA results on both closed-set and open-set datasets.

**Comparison with Regularization Techniques.** To further validate the robustness of our DCM, we compare it against several current regularization techniques by integrating it with the SOTA robust loss function (*i.e.*, ANL) and sample selection (*i.e.*, SED) methods. As illustrated in Table 3, these popular methods cause performance degradation when combined with SOTA methods. In contrast, our DCM consistently improves noise robustness across varying noise rates. From this observation, random neuron or connection dropping strategies such as Dropout and DropConnect fail to provide additional benefits for existing noise-robust methods. In contrast, by selectively processing information, our DCM can be adapted to existing methods. Compared with CDR, our method offers a more effective criterion for critical information selection. Additionally, CDR's identification of important parameters depends on gradient computation, which is comparatively

Table 3: Comparison with popular regularization techniques on the CIFAR-100 dataset across various noise rates. The average test accuracy (%) is reported over the last 10 epochs.

| Methods | Sym-0.2 | Sym-0.4 | Sym-0.6 | Sym-0.8 | Asym-0.2 |
|---|---|---|---|---|---|
| *Combining with Robust Loss Function* | | | | | |
| ANL | 67.05 | 62.02 | 51.78 | 28.01 | 66.24 |
| ANL-Dropout | 66.65(-0.40) | 61.64(-0.38) | 50.95(-0.83) | 27.53(-0.48) | 65.86(-0.38) |
| ANL-DropConnect | 66.09(-0.96) | 59.66(-2.36) | 45.10(-6.68) | 18.48(-9.53) | 64.99(-1.25) |
| ANL-CDR | 67.06(+0.01) | 60.78(-1.24) | 49.13(-2.65) | 15.61(-12.40) | 64.51(-1.73) |
| **ANL-DFC (ours)** | **67.71**(+0.66) | **62.88**(+0.86) | **52.65**(+0.87) | **30.52**(+2.51) | **66.83**(+0.59) |
| *Combining with Sample Selection* | | | | | |
| SED | 66.50 | 64.52 | 59.29 | 38.15 | 66.39 |
| SED-Dropout | 63.74(-2.76) | 61.61(-2.91) | 57.21(-2.08) | 37.20(-0.95) | 63.97(-2.42) |
| SED-DropConnect | 63.26(-3.24) | 61.62(-2.90) | 56.66(-2.63) | 36.77(-1.38) | 63.18(-3.21) |
| SED-CDR | 66.62(+0.12) | 63.36(-1.16) | 58.57(-0.72) | 38.49(+0.34) | 65.63(-0.76) |
| **SED-DFC (ours)** | **66.83**(+0.33) | **64.77**(+0.25) | **60.01**(+0.72) | **39.18**(+1.03) | **66.78**(+0.39) |

Table 4: Comparison with the SOTA methods on Webvision-Mini. The Top-1 validation accuracy(%) is reported, and the top 2 best results are highlighted using boldface and underlining.

| Method | Publication | Accuracy (%) |
|---|---|---|
| Decoupling | NeurIPS 2017 | 62.54 |
| D2L | ICML 2019 | 62.68 |
| MentorNet | ICML 2018 | 63.00 |
| Co-teaching | NeurIPS 2018 | 63.58 |
| INCV | ICML 2019 | 65.24 |
| ELR+ | NeurIPS 2020 | 77.78 |
| GJS | NeurIPS 2021 | 77.99 |
| CC | ECCV 2022 | 79.36 |
| DISC | CVPR 2023 | 80.28 |
| DISC-DFC | Ours | 80.80 |
| **DISC-DKAN** | | **81.00** |

Table 5: Comparison with the SOTA methods on Clothing1M. The results with * are reimplemented using open-source code, and others are directly from the original paper.

| Method | Publication | Accuracy (%) |
|---|---|---|
| Co-teaching | NeurIPS 2018 | 69.21 |
| JoCoR | CVPR 2020 | 70.30 |
| DMI | NeurIPS 2019 | 72.46 |
| ELR+ | NeurIPS 2020 | 74.39 |
| GJS | NeurIPS 2021 | 71.64 |
| CAL | CVPR 2021 | 74.17 |
| DISC | CVPR 2023 | 73.72 |
| SURE* | CVPR 2024 | 72.57 |
| SURE-DFC | | 73.39 |
| DISC-DFC | Ours | 74.15 |
| **DISC-DKAN** | | **74.49** |

less efficient. Consequently, our DCM possesses greater generalizability, allowing for direct plug-and-play integration with existing methods.

### 4.3 EVALUATION ON REAL-WORLD DATASETS

The experimental results on WebVision-Mini and Clothing1M are shown in Table 4 and 5, respectively. Specifically, our DISC-KAN achieves SOTA performance with accuracy of 81.00% and 74.49%, respectively. Furthermore, our method maintains an accuracy advantage of 0.82% over the SURE on Clothing1M. Due to incompatibility between SURE's cosine classifier and KAN's continuous spline representations, we implement only SURE-DFC. The results highlight that KAN exhibits superior noise robustness over FC in real-world scenarios. Comprehensive evaluations on both synthetic and real-world datasets verify that our DCM offers plug-and-play compatibility with existing methods while consistently achieving SOTA performance.

### 4.4 ROBUSTNESS ANALYSIS

**Gradient Error Analysis**. We provide a gradient error analysis to validate our DCM's capability to suppress the gradient backpropagation from noisy labels. Let $\mathcal{L}(f(x;\theta), y)$ and $\tilde{\mathcal{L}}(f(x;\theta), \tilde{y})$ denote the loss function supervised by clean and noisy labels, respectively. Then, the gradient of the visual backbone parameters $\theta_\psi$ under noisy labels is given by: $\nabla_{\theta_\psi} \tilde{\mathcal{L}} = \sum_{i=1}^{B} \frac{\partial \tilde{\mathcal{L}}(f(x_i;\theta), \tilde{y}_i)}{\partial \theta_\psi}$. To quantify gradient error $\varepsilon_f$ caused by label noise with the model $f$, we define it as the discrepancy between

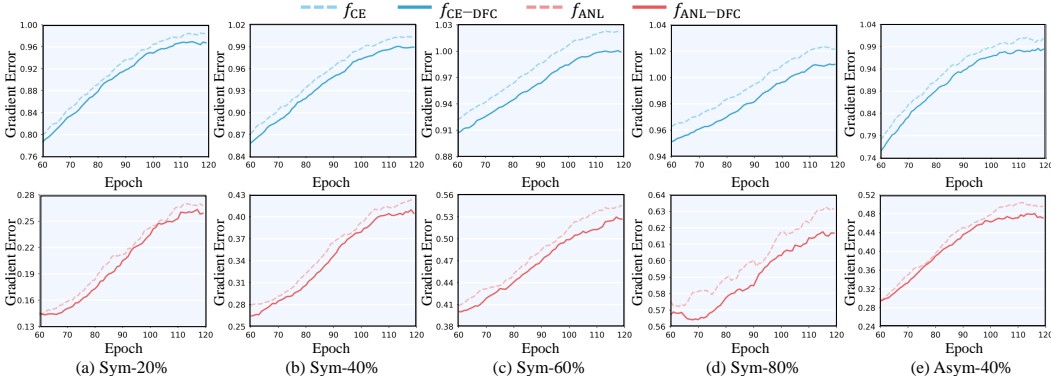

Figure 3: Comparison of gradient error $\varepsilon_f$ across different models under various noise levels on the CIFAR-10 dataset. Specifically, $f_{\text{CE-DFC}}$ and $f_{\text{ANL-DFC}}$ denote the classifier with our dynamic connection masking combined with CE and ANL, respectively. The $f_{\text{CE}}$ and $f_{\text{ANL}}$ represent the original CE and ANL methods with a FC classifier. Figures (a) to (e) illustrate the average cosine similarity between clean and noisy gradients of the last layer backbone parameters over epochs.

the gradients induced by clean and noisy labels:

$$\varepsilon_f = 1 - \cos(\nabla_{\theta_\psi}\mathcal{L}, \nabla_{\theta_\psi}\tilde{\mathcal{L}}) = 1 - \frac{\nabla_{\theta_\psi}\mathcal{L} \cdot \nabla_{\theta_\psi}\tilde{\mathcal{L}}}{\left\|\nabla_{\theta_\psi}\mathcal{L}\right\|_2 \times \left\|\nabla_{\theta_\psi}\tilde{\mathcal{L}}\right\|_2}, \quad (5)$$

where $\nabla_{\theta_\psi}\mathcal{L}$ represents the optimal gradient, and $\nabla_{\theta_\psi}\tilde{\mathcal{L}}$ denotes the noise-corrupted optimization. We utilize the cosine similarity to quantify the consistency between the noisy and clean gradients ($\nabla_{\theta_\psi}\mathcal{L}$ and $\nabla_{\theta_\psi}\tilde{\mathcal{L}}$). A higher cosine similarity indicates that the noise-corrupted gradient $\nabla_{\theta_\psi}\tilde{\mathcal{L}}$ approximates the optimal gradient $\nabla_{\theta_\psi}\mathcal{L}$, resulting in a smaller gradient error $\varepsilon_f$ and demonstrating greater robustness to label noise. To present an intuitive analysis, we compute the gradient error $\varepsilon_f$ of the backbone's final layer parameters during training. This comparative analysis is conducted using both CE and ANL loss functions with different classifiers. Specifically, the model performs the backpropagation using both clean and noisy labels to obtain the corresponding gradients $\nabla_{\theta_\psi}\mathcal{L}$ and $\nabla_{\theta_\psi}\tilde{\mathcal{L}}$ for recording. Only the noisy gradients $\nabla_{\theta_\psi}\tilde{\mathcal{L}}$ are utilized for parameter updating. As illustrated in Figure 3, as the noise rate increases, our $f_{\text{CE-DFC}}$ and $f_{\text{ANL-DFC}}$ models consistently yield lower gradient error. This empirical evidence indicates that our approach reduces the gradient error $\varepsilon_f$ under noisy supervision, thereby mitigating the adverse effects of noisy labels.

**Confidence Analysis.** We conduct a confidence analysis to evaluate the model's fitting degree to clean and noisy data throughout training, which can be measured by their respective average prediction probabilities. Specifically, we define the average prediction probability on clean and noisy labels as their corresponding confidences. As illustrated in Figure 4, we visualize the noisy and clean confidences during training for CE with different classifiers on CIAFR10. It can be observed that our $f_{\text{CE-DFC}}$ and $f_{\text{CE-KAN}}$ exhibit lower noisy confidence across various noise levels, demonstrating the efficacy of our DCM in mitigating overfitting to noisy data.

In contrast, our $f_{\text{CE-DFC}}$ and $f_{\text{CE-KAN}}$ maintain comparable clean confidence to their fully connected counterparts under various noise levels. Notably, even under high-noise conditions (Figure 4(d)), the model with our DCM achieves a superior fitting to the clean samples. The phenomenon validates that our DCM can adequately fit clean samples under noisy conditions, thereby ensuring the model's fitting capability. To conclude, our DCM effectively mitigates overfitting to noisy data without degrading the clean information.

### 4.5 ABLATION STUDY

**Masking method analysis.** We implement weight and edge-wise masking strategies to evaluate the efficiency of our method, where former masks edges by sorting weights and the latter globally discards unimportant edges. As shown in Table 6, both node-wise and edge-wise masking methods

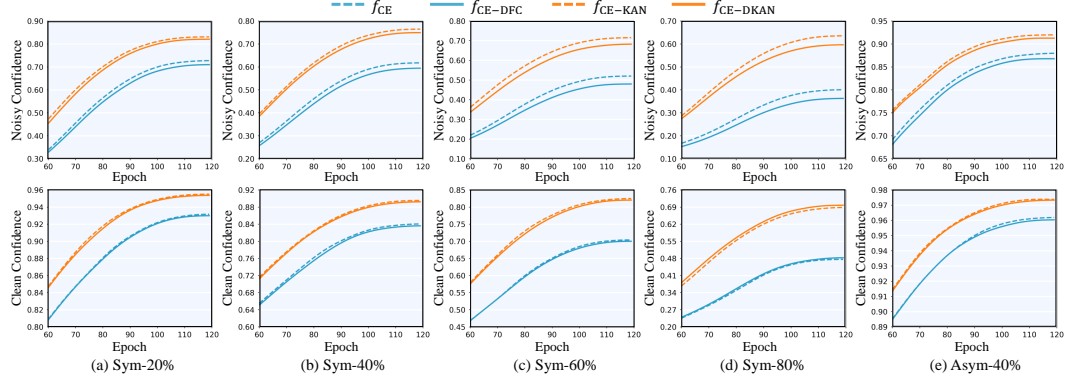

Figure 4: Noisy and clean confidence analysis across different classifiers on CIFAR-10. To facilitate clear comparison, results from mid-training to final epochs are presented.

Table 6: Comparison of various masking methods on CIFAR-10 with $80\%$ symmetric noise. Both the highest (Best) and average test accuracy (Avg.) over the last 10 epochs are reported.

| Method | Masking Method | Best | Avg. |
|---|---|---|---|
| ANL | - | 61.45 | 61.33 |
| ANL-DFC | By Weight | 63.57 | 62.96 |
|  | Edge-wise | 63.91 | 62.92 |
|  | **Node-wise** | **65.19** | **63.93** |
| ANL-KAN | - | 60.56 | 60.22 |
| ANL-DKAN | Edge-wise | 62.64 | 61.54 |
|  | **Node-wise** | **64.04** | **63.63** |

Table 7: Comparison with different masking stages on CIFAR-10 with 80% symmetric noise. We report the highest (Best) and the average test accuracy (Avg.) over the last 10 epochs.

| Method | Masking Stage | | Accuracy (%) | |
|---|---|---|---|---|
|  | Training | Testing | Best | Avg. |
| CE-DFC |  |  | 39.01 | 18.62 |
|  | ✓ |  | **41.82** | **19.65** |
|  | ✓ | ✓ | 40.56 | 18.80 |
| CE-DKAN |  |  | 41.48 | 17.62 |
|  | ✓ |  | **42.74** | **18.08** |
|  | ✓ | ✓ | 41.80 | 17.87 |

outperform directly masking edges with the lowest weights. This improvement stems from our method's simultaneous consideration of feature information and edge weights, thereby effectively maintaining critical connections. Furthermore, node-wise masking demonstrates greater robustness than edge-wise masking, as edge-wise masking may cause more significant architectural changes, while node-wise masking preserves training capability for all nodes. Consequently, we adopt node-wise masking, which better maintains critical network topology and improves noise robustness.

**Masking stage analysis.** We investigate the effect of applying DCM during different stages under $80\%$ symmetric noise on CIFAR-10. As demonstrated in Table 7, the results indicate that employing DCM solely during the training phase yields better performance than applying it in both training and testing phases. This result demonstrates that DCM primarily serves as an effective regularization technique, hindering the model's overfitting to noisy data. Consequently, in this study, we adopt the strategy of applying DCM exclusively during the training phase to enhance noise robustness.

## 5 CONCLUSION

In this study, we propose a novel dynamic connection masking (DCM) mechanism for the widely-used FC-based classifiers to combat noisy labels. Our DCM approach can adaptively mask unimportant edges during training while preserving the most informative pathways. Through robustness analysis, we demonstrate that our DCM effectively mitigates gradient errors propagated from noisy labels while simultaneously maintaining its capacity to fit clean samples. Additionally, our DCM is also compatible with the newly-emerged Kolmogorov-Arnold Network (KAN) architecture, effectively boosting its robustness against noisy labels. Comprehensive experiments integrating DCM with various noise-robust training methods across synthetic and real-world datasets consistently validate the effectiveness of our approach in noisy label learning scenarios.

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
