# OpenReview forum: "Combating Noisy Labels via Dynamic Connection Masking"
_ICLR.cc/2026/Conference — ICLR 2026 Conference Withdrawn Submission_

### Official Review · Reviewer_5jjv · 2025-10-22

**Soundness:** 4
**Presentation:** 4
**Contribution:** 3
**Rating:** 4
**Confidence:** 4

**Summary:**

The paper proposed a new method `Dynamic Connection Masking (DCM)` technique that can be applied on fully connected layer of a neural net. The proposed mechanism adaptively masks the less important edges based on the noisy labels present in the training set. The proposed method also can be be applied on KAN architecture. The results demonstrated better or on-par SOTA performance.

**Strengths:**

- The paper is written nice and the presentation is clear for following along the method and the results.
- The paper proposes a simple and robust technique that handles noisy labels in real world datasets.
- The technique is scalable since the proposed network regularizations are applied on the FC layer of the network.
- A nice novel network regularization method as opposed to the dominant SOTA world with loss functions to deal with noisy labeled data.

**Weaknesses:**

- Dropout kind of SOTA methods generally not applied during inference time, for DCM, it is unclear if it is applied during inference or not.
- The edge importance scoring (Eq. 1-- 2) and masking (Eq 3 -- 4) formulations are easy to understand when applied on an FC layer and the formulations are presented with that use case. However, it is unclear how these formulations will transform when applied for a KAN.
- There is enough evidence for synthetic noise and a couple of real-world noise benchmarks, however, the experiments lack evidence for out-of-distribution test sets.
- Lacks an analysis on the computational overhead introduced by the new variables *W*, *A*, *U*
- The experimental results does not include the performance of the DCM alone, it always is combined with some SOTA method, APL, ANL, SED, DISC or others.

**Questions:**

- Clarify how the proposed DCM formulations can be transformed or applied to the KAN architecture. In case of KAN, is it applied on all learned functions or only a few? Provide those conceptual clarifications in the proposed methodology
- Provide clarification on if DCM i) can be applied in the inference time or not? ii) if not please clarify since that part is not clear from the paper.
- On a related note, if the method is scalable and also robust as claimed in the gradient error analysis, demonstrate the efficacy of DCM on one or two OOD benchmarks with/without enabling DCM during inference time (whichever is apt)
- What is the performance of DCM alone either on FC or KAN? It is always combined with some SOTA method, it is unclear to understand the real gains of the proposed method. Are the gains in performance coming from the SOTA methods or the DCM or the combination of both. Please provide that evidence and clarify the real benefits of the method.

---

### Official Review · Reviewer_Q6Zf · 2025-10-30

**Soundness:** 2
**Presentation:** 3
**Contribution:** 3
**Rating:** 4
**Confidence:** 4

**Summary:**

The authors propose a weight pruning based approach to address the LNL problem.
The core idea is simple yet effective: by computing the batch-wise standard deviation of activations in the final layer (prior to the summation in the inner product), the method identifies and removes less informative parameters.
This pruning operation alone enables the model to learn robustly against label noise, and when combined with existing LNL methods, it yields a clear and consistent improvement in performance.

**Strengths:**

The proposed method is novel. While selective parameter updates have been explored in fields such as multi-task learning and continual learning, to the best of my knowledge, this is the first work to apply such a mechanism within the LNL domain.
Moreover, leveraging the batch-level standard deviation to measure activation importance is a highly reasonable design choice that aligns well with the noise characteristics of LNL settings.
Although some of the experimental improvements are modest, the paper presents a fresh and meaningful direction for future research in robust learning under noisy supervision.

**Weaknesses:**

The introduction of Kolmogorov–Arnold Networks (KAN) is intriguing, but its connection to the main contribution of the paper is not clearly articulated. It remains unclear whether integrating DCM with KAN provides additional benefits or merely serves as a separate demonstration. If it is the former, an ablation study isolating the effects of KAN would substantially strengthen the argument.

Additionally, the proposed approach appears highly sensitive to batch size. Since the method computes importance scores based on batch-wise standard deviation, smaller batches could lead to unstable variance estimation. It would be valuable to include experiments showing the method’s behavior under varying batch sizes, or at least to provide practical guidelines for selecting an appropriate batch size.

**Questions:**

If the authors can address some of the concerns mentioned in the Weaknesses section, I believe this paper would be sufficiently strong to be accepted at ICLR and make a meaningful contribution to the community.

In particular, providing stronger evidence through additional analyses such as the integration of KAN and ablation studies that clarify the strengths and limitations of the proposed method would significantly enhance the overall impact of the paper.

---

### Official Review · Reviewer_oaWk · 2025-11-01

**Soundness:** 1
**Presentation:** 2
**Contribution:** 1
**Rating:** 2
**Confidence:** 4

**Summary:**

The authors propose Dynamic Connection Masking (DCM), a mechanism that selectively masks less important edges in fully connected layers (FCN) to improve robustness against label noise. The effectiveness of DCM is validated on both synthetic and real-world benchmarks, and further evaluated on Kolmogorov–Arnold Networks (KANs).

**Strengths:**

- The paper is clearly written and easy to follow.

**Weaknesses:**

- Lack of theoretical motivation: There is neither theoretical analysis nor empirical validation to justify the claimed relationship between edge importance scoring and noisy-label learning.

- Lack of empirical investigation: The authors should conduct experiments under normal training settings without applying DCM, record the edge importance scores throughout training, and validate whether the edges that DCM intends to mask are connected to noisy labels.

- Novelty concern: The idea of assigning importance scores to edges is not novel and is closely related to the CDR (Xia et al., 2021) method, as acknowledged by the authors. However, the comparison with CDR lacks sufficient evidence. The justification based on computational overhead is weak, since the additional cost of CDR is negligible and it reuses gradients from the standard update step rather than introducing new gradient computations.

- Insufficient experimental differentiation: The paper should include experiments that visualize or quantitatively compare the mask selections of DCM and CDR. Demonstrating how the two methods differ in their selected edges is essential to clarify the contribution and originality of the proposed method.

- Experimental gains are not sufficient: In most cases, the improvements are less than 1%, which is not convincing for a label-noise task, as different random seeds can significantly affect sampling and final performance.

References:
- Xiaobo Xia, Tongliang Liu, Bo Han, Chen Gong, Nannan Wang, Zongyuan Ge, and Yi Chang. Robust early-learning: Hindering the memorization of noisy labels. International Conference on Learning Representations (ICLR), 2021.

**Questions:**

Please refer to the weaknesses listed above.

---

### Official Review · Reviewer_xjV6 · 2025-11-01

**Soundness:** 3
**Presentation:** 3
**Contribution:** 3
**Rating:** 4
**Confidence:** 4

**Summary:**

This paper proposes Dynamic Connection Masking (DCM), a novel architectural regularization mechanism designed to improve model robustness under noisy labels. DCM dynamically masks less informative edges in fully connected layers based on their information-carrying capacity, thereby reducing gradient noise propagation during training. Experiments desmonstrate its effectiveness.

**Strengths:**

1. This paper proposed a novel perspective from model architecture for mitigating label noise singal.
2. The paper is well-written and clearly presented.
3. The method design sounds resaonable.

**Weaknesses:**

1. Instead of KAN network, experiments should introduce extra trials on CLIP (the transformer structure based network).
2. More ablation studies are required.
 - In equation 2, the importance score is computed among the batch scale. Whether a larger value of $B$ is important for accurate estimation and stable training?
 -  The selection of hyper-parameter $p$.
3. Why adopting both training and testing mask will lead to worse performance compared to only training mask? Intuitively, the parameter masked in training phase should be masked for consistent test.
4. The compared methods are out-of-fashion. To my knoeledge, existing sample-selection based methods achieved better performance reported in the paper. I suggest the author to add their methods to the SOTA sample-selection methods (for example, SelectMix) or directly compare with them.

**Questions:**

See weaknesses.

---

### Note · Authors · 2025-11-14

I have read and agree with the venue's withdrawal policy on behalf of myself and my co-authors.